# Mind the Gap: Assessing Temporal Generalization in Neural Language Models

**Angeliki Lazaridou**$^{*\heartsuit\triangle\spadesuit}$ **Adhiguna Kuncoro**$^{\star\heartsuit\triangle}$ **Elena Gribovskaya**$^{\star\heartsuit\triangle}$
**Devang Agrawal**$^{\diamond\heartsuit}$ **Adam Liška**$^{\diamond\heartsuit}$ **Tayfun Terzi**$^{\diamond}$ **Mai Gimenez**$^{\diamond}$
**Cyprien de Masson d'Autume**$^{\diamond}$ **Tomas Kocisky**$^{\heartsuit}$ **Sebastian Ruder**$^{\heartsuit}$
**Dani Yogatama**$^{\clubsuit}$ **Kris Cao**$^{\clubsuit}$ **Susannah Young**$^{\clubsuit}$ **Phil Blunsom**$^{\clubsuit\spadesuit}$
DeepMind, London, UK
{angeliki,akuncoro,egribovskaya}@deepmind.com

## Abstract

Our world is open-ended, non-stationary, and constantly evolving; thus what we talk about and how we talk about it change over time. This inherent dynamic nature of language contrasts with the current static language modelling paradigm, which trains and evaluates models on utterances from overlapping time periods. Despite impressive recent progress, we demonstrate that Transformer-XL language models perform worse in the realistic setup of predicting future utterances from beyond their training period, and that model performance becomes increasingly worse with time. We find that, while increasing model size alone—a key driver behind recent progress—does not solve this problem, having models that continually update their knowledge with new information can indeed mitigate this performance degradation over time. Hence, given the compilation of ever-larger language modelling datasets, combined with the growing list of language-model-based NLP applications that require up-to-date factual knowledge about the world, we argue that now is the right time to rethink the static way in which we currently train and evaluate our language models, and develop *adaptive* language models that can remain up-to-date with respect to our ever-changing and non-stationary world. We publicly release our dynamic, streaming language modelling benchmarks for **WMT** and **ARXIV** to facilitate language model evaluation that takes temporal dynamics into account.[1]

## 1 Introduction

In recent years, substantial efforts in neural language modelling have focused on finding better neural architectures, building increasingly larger models, and compiling ever-larger amounts of training data, which have been shown to endow language models with the ability to perform well on a wide variety of downstream tasks with minimal fine-tuning (Vaswani et al., 2017; Radford et al., 2019; Brown et al., 2020). While this approach has led to impressive progress, it nevertheless relies on a static experimental paradigm. Concretely, the prevailing practice is to curate a large pretraining web crawl—randomly partitioned into a training set and a validation set in a time-agnostic fashion—and then evaluate on tasks and benchmarks that mostly overlap in time with the pretraining data.[2]

---

$^{*}$Equal contribution. $^{\spadesuit}$ Project initiation. $^{\triangle}$ Paper writing. $^{\diamond}$ Project technical infrastructure. $^{\heartsuit}$ Model design and experiments. $^{\clubsuit}$ Project support and advice.

[1]We release our dynamic (streaming) language modelling benchmark for **WMT** and **ARXIV** at `https://github.com/deepmind/deepmind-research/tree/master/pitfalls_static_language_models`.

[2]In the case of GPT-3 (Brown et al., 2020), such tasks include LAMBADA (Paperno et al., 2016), TriviaQA (Joshi et al., 2017b), and WMT translation datasets, among others. These tasks were introduced between 2014 and 2017, which overlap in time with the GPT-3 CommonCrawl dataset that covered the period of 2016-2019.

35th Conference on Neural Information Processing Systems (NeurIPS 2021)

In this work, we argue that such practices carry two potential risks. First, they do not assess a language model's ability to generalize well to future data from beyond their training period—an important ability we henceforth refer to as temporal generalization. In our dynamic and non-stationary world, temporal generalization is a key necessity: Many practical machine learning systems that use language model (LM) pretraining, such as machine translation and dialogue systems, are deployed on utterances that users will say in the future, whilst being trained on utterances that users have already said in the past. Furthermore, temporal generalization is also crucial to perform well on realistic use cases of language models in the real world. Examples include flagging fake news about recent events that happen outside of the training period (Thorne and Vlachos, 2018; Zellers et al., 2019; Augenstein et al., 2019), forecasting stock prices from the latest news articles (Ding et al., 2015), and answering knowledge-intensive questions like "How many people have been infected by COVID-19?" and "Has the USA ever had a female Vice President?", whose answers have evolved with time.

Second, the temporal overlap between the training and evaluation data increases the risk of "test data contamination", where parts of the evaluation task are unwittingly included in the pretraining data. Indeed, many language modelling evaluations treat the data as independent and identically distributed (i.i.d) at either the sentence (Chelba et al., 2013) or document level (Brown et al., 2020; Gao et al., 2021). Nevertheless, language modelling data are not i.i.d. (neither at the word, sentence, or document level); rather it is a time series, and thus models trained on the prefix of a sample from the series should be evaluated on the continuation of that series. While previous research (Levenberg et al., 2010) has highlighted the importance of temporal splits for fairer and more realistic evaluations—and has led to research (Osborne et al., 2014; Yogatama et al., 2014) that addresses language modelling from this streaming perspective (§7)—using temporal splits (or splits beyond random ones) is still the exception rather than the rule, as evidenced by many contemporary LM (Brown et al., 2020; Gao et al., 2021) and downstream tasks (Lewis et al., 2020a) that are affected by test data contamination.[3]

Here we begin with our first question: To what extent does the current static language modelling practice overestimate performance, compared to the more realistic setup that evaluates LMs on future utterances? To this end, we introduce our dynamic, streaming language modelling benchmarks (§2), and find that Transformer-XLs (Dai et al., 2019) perform up to 16% worse when predicting articles that are published up to 2 years after the end of the training period. Moreover, model performance becomes increasingly worse with time (§3). Given this finding, we ask: What kinds of predictions is the model struggling with in the dynamic evaluation setup?—-which we answer in §3.1.

Beyond LM perplexity evaluation, we further ask: How exactly does this temporal performance degradation of Transformer LMs manifest in different types of question-answering (QA) tasks? We answer this through two different QA tasks, including one around recent events happening outside of the LM training period (§5). Lastly, given the challenges presented by temporal generalization for LMs: What, then, is the remedy? This question is important because keeping LMs up-to-date by retraining with new data is expensive in compute and carbon costs (Strubell et al., 2019; Patterson et al., 2021), and risks the model getting outdated in-between long retraining cycles.[4] We find that increasing model size alone—a key driver behind recent LM progress (Kaplan et al., 2020)—is not a solution for the temporal generalization problem (§4): Larger models suffer from the same performance degradation with time, and a smaller model trained on more recent data can outperform a 60% larger model that lacks access to more recent data. We then explore a simple yet effective way of keeping our models up-to-date by continually updating the model's parameters through dynamic evaluation (Mikolov et al., 2010; Krause et al., 2019), which performs a few steps of gradient descent on streams of new data (§6), and outline other promising approaches in this direction (§7). We conclude with the following recommendations for future LM research:

- We should evaluate LMs on their generalization ability to future data, which circumvents test data contamination, rewards models that generalize beyond the surface patterns of their pretraining data, and better reflects how large LMs are used in practical systems. We thus argue for the broader inclusion of timestamp information in pretraining data and downstream tasks to make this possible.
- Stale LMs that are deployed far outside of their training period perform substantially worse on downstream tasks that require up-to-date factual knowledge, although a broader set of experiments are needed to pinpoint what kinds of tasks are most affected. Our findings also highlight the need

---

[3]Brown et al. (2020) used $n$-gram filtering and deduplication to remove overlaps between the training and test sets. This can potentially induce a correlation between the training and evaluation sets that LMs can exploit.

[4]These risks are exacerbated by the trend of ever-larger LMs, where retraining incurs even higher costs.

| Dataset | Domain | Time period | #Words per Doc (Average) | Training Size (in GB) | Prop. of CONTROL's Training Data from the Test Period |
|---|---|---|---|---|---|
| **WMT** | News | 2007 - 2019 | 551 | 22.65 | 6.3% |
| **CUSTOMNEWS** | News | 1969 - 2019 | 491 | 395.59 | 34.8% |
| **ARXIV** | Scientific text | 1986 - 2019 | 172 | 0.72 | 14.5% |

Table 1: Statistics and time periods of the datasets used in this study.

for more tasks, benchmarks, and metrics that evaluate *how well* and *how rapidly* LMs are able to integrate new information, which are important ingredients to encourage progress in this direction.

- All in all, above and beyond impressive scaling efforts towards ever-larger models (Brown et al., 2020; Fedus et al., 2021), we argue for the development of *adaptive* language models that can remain up-to-date with respect to our open-ended and non-stationary world.

## 2    Time-stratified language modelling

We begin by introducing our time-stratification experimental setup, which examines *how well* Transformer LMs perform when evaluted on future utterances from beyond their training period.

### 2.1    Datasets

We identify news and scientific articles as two sources of dynamic streaming data with a naturally changing distribution over time—lending themselves well to evaluating how well language models generalize over time. For the scientific domain, we use the publicly available arXiv abstracts (**ARXIV**).[5] For news, we use the publicly available WMT News Crawl (**WMT**).[5] We ensure that any trends we observe also generalize well to models trained on larger datasets—which reliably improve language modelling and downstream task performance (Liu et al., 2019)—by compiling a larger news corpus that we term **CUSTOMNEWS**. This dataset consists of crawled English news sources from 1969-2019, and covers various topics including politics, finance, and sport. We apply minimal preprocessing through: (i) Removal of non-English documents, (ii) deduplication using the MinHash algorithm, and (iii) tokenization using Moses.[5] Table 1 summarizes key statistics of our datasets.

### 2.2    Experiment: A model up to 2 years stale

**Evaluation period and test set.**    For each dataset, we pick the last two years (i.e. 2018 and 2019) as our evaluation period, and sub-sample a test set of 24k test documents (1k per test month).

**TIME-STRATIFIED setup.**    In this setup, we evaluate LMs trained on the past based on their ability to predict future articles that are published after the time period of their training data; this split is constructed using the time stamp of each article. Here we use all documents from the beginning of each dataset's time period up until September 2017 as training data, and use the last three months of 2017 as our validation period; we denote this as the **TIME-STRATIFIED** setup. We then evaluate the model on the 2018-2019 test set above, which evaluates the model's ability to generalize across time by predicting articles up to two years after the end of their training period—a realistic time frame during which we expect large-scale language models to be used without retraining on recent data.

**CONTROL setup.**    We assess whether time stratification poses a challenge for current LMs by comparing it with the following **CONTROL** setup. In this setup, the training set includes documents that come from the same 2018-2019 period as the evaluation set (naturally excluding the test documents themselves). This CONTROL setup thus resembles the prevailing (static) language modelling experimental practices, which train and evaluate LMs on text data from overlapping time periods.

Crucially, we control such that the two training sets are of the exact same size, i.e., they differ only in the time periods of their training data, rather than in their absolute training set sizes. Here we construct the CONTROL training data by taking the most recent documents starting from the end of the evaluation period (excluding the test documents and including the same number of training documents per test month), and keep adding documents from previous time periods until we reach the same training size as the TIME-STRATIFIED setup. In Table 1, we report the proportion of documents in the

---

[5]ArXiv:    `https://arxiv.org/help/oa/index`;    WMT    News:    `http://data.statmt.org/news-crawl`; and SacreMoses: `https://github.com/alvations/sacremoses`.

CONTROL setup's training data that come from the same 2018-2019 time period as the evaluation set, which is higher for ARXIV and CUSTOMNEWS due to their recent exponential growth of new documents. We sample a similarly-sized validation set as the TIME-STRATIFIED setup, which in this case comes from the 2018-2019 evaluation period (again excluding the test documents). Importantly, both the TIME-STRATIFIED and CONTROL models are evaluated on the exact same test set from the 2018-2019 period, which facilitates a fair perplexity comparison between the two setups.

**Relative perplexity comparison.** We want to measure temporal degradation, i.e. do Transformer LMs perform increasingly worse when predicting test documents further into the future? However, any *absolute* perplexity degradation of the TIME-STRATIFIED model over time (e.g., perplexity for Jan. 2018 vs Dec. 2018) is an unreliable measure: Some months have longer documents, which lead to higher perplexity. We thus measure temporal degradation through *relative* perplexity changes between the TIME-STRATIFIED and CONTROL models for the same test month (e.g. Dec. 2018).

### 2.3 Model

We perform our experiments on autoregressive, left-to-right LMs. We use a Transformer-XL (Dai et al., 2019) with 18 layers and 1,024 hidden units, resulting in 287M parameters—roughly 15% smaller than GPT-2$_{\text{MEDIUM}}$ and BERT$_{\text{LARGE}}$; we later explore larger models in §4. We set the Transformer sequence length to 1,024, and set the memory cache length to 384 during training and 1,600 during test. We use a vocabulary of 50,259 subwords, obtained via SentencePiece (Kudo and Richardson, 2018) trained on a random subset (up to 15GB) of the training data of each respective experiment, i.e., CONTROL and TIME-STRATIFIED. Training and validation are done on subword tokens, but to facilitate our later analysis (§3.1), all test perplexities are computed over actual test word tokens,[6] whose negative log probabilities are obtained by summing those of their subwords.

## 3 Language Modelling Experiments & Analysis

**To what extent does the static CONTROL setup overestimate model performance, compared to the more realistic TIME-STRATIFIED setup that evaluates LMs on future utterances?** Figure 2 presents the results of our first experiment. Although we train both models: (i) On the exact same dataset sizes, and (ii) using the same model architectures, a stale TIME-STRATIFIED model performs worse than the

| Setup | WMT | CUSTOM NEWS | ARXIV |
|---|---|---|---|
| CONTROL | 21.11 | 18.38 | 21.38 |
| TIME-STRATIFIED | 22.45 | 21.33 | 23.07 |
| Δ, absolute | +1.34 | +2.95 | +1.69 |
| Δ, relative (%) | 6.34 | 16.04 | 7.90 |

Table 2: Perplexity of Transformer-XL when trained with the two different setups, and evaluated on the same test set from the 2018-2019 period.

CONTROL model, which *has* seen training data from the test period—with up to 16% perplexity difference. We attribute the higher relative degradation on CUSTOMNEWS and ARXIV to their recent exponential growth of new documents, resulting in a higher proportion of documents from the test period in the data (Table 1), hence presenting a more difficult temporal generalization problem.

**Do Transformer LMs perform increasingly worse when predicting future utterances further away from their training period?** To this end, Fig. 1 plots the relative perplexity increase of the TIME-STRATIFIED over the CONTROL model. As evidenced by the upward slope on all datasets, the model deteriorates more as we ask it to predict data further away from the training period, affirming that the model indeed becomes *increasingly outdated* with time. **How general are these findings?** We find that the same patterns not only generalize across datasets, as we have just shown, but are also found: (i) For test years other than 2018-2019 (Appendix A.1), (ii) beyond the two-year tem-

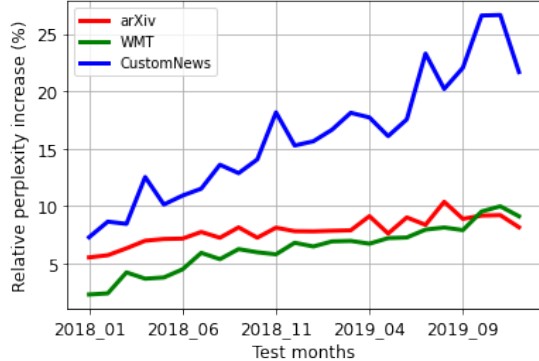

Figure 1: Relative ppl. increase of TIME-STRATIFIED over CONTROL, across test months.

---

[6] An example is detokenizing "__contact", "less" into "contactless", where "__" denotes a token boundary.

poral gap between the end of the training and test periods (Appendix A.2), and (iii) across other languages (German WMT, Appendix A.3).

## 3.1 Analysis

Having established that model performance degrades with time, we now turn to investigate the following question: What exactly are the kinds of predictions that the model is struggling with?

**Part-of-speech (POS) tag breakdown.** We present the relative perplexity increase of the TIME-STRATIFIED over the CONTROL model, broken down by POS tag and across time (Fig. 2, solid lines). First, we see that performance on common nouns (orange line), the most frequent POS tag, degrades with time; in fact, performance degradation on common nouns drives the overall degradation trend (brown line). Moreover, the TIME-STRATIFIED model's performance degrades most rapidly when making temporal generalizations about proper nouns (blue line) and numbers (purple line). Qualitative analysis indicates that the model performs badly on named entities in politics, whose position changed during our 2018-2019 evaluation period (e.g., "Bolsonaro", "Pompeo", "Khashoggi"). This degradation is consequential because proper nouns—and by extension named entities—closely relate to up-to-date factual world knowledge; in §5 we explore how exactly this degradation affects different downstream tasks. Interestingly, we also found the model struggling with concepts associated with cultural and sociological changes on which public perception and discourse have evolved over time, such as "MeToo" and "BlackLivesMatter" (Bender et al., 2021).

**Perplexity and topics.** We analyze *how* the speed of the TIME-STRATIFIED model's perplexity degradation relates to different topics. We first cluster the documents using Latent Dirichlet Allocation (Blei et al., 2003, LDA), which represents each document as a mixture of topics and each topic as a distribution over words; we then aggregate the perplexity of words in the test documents by topic. We observe that model performance on topics around politics and sports change more rapidly with time than topics around lifestyle, as shown in (Fig. 2, shown in the three dotted lines).

**Perplexity and temporal frequency shifts** In practice, *adaptation* is a key necessity to maximize the potential of LMs in our dynamic and non-stationary world. This includes the ability to integrate information about new words and concepts that *never occurred* in the past, and also words whose context or meaning *had sub-*

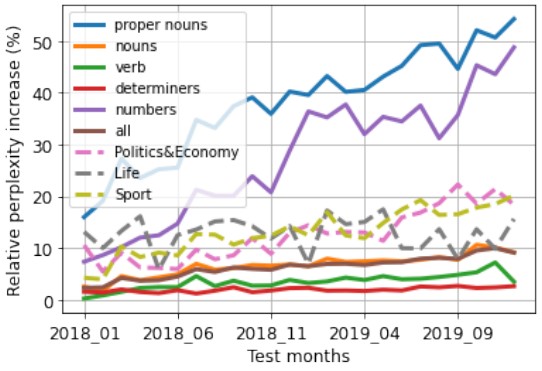

Figure 2: **WMT** relative ppl. increase of the TIME-STRATIFIED over the CONTROL models, broken down by part-of-speech (POS) tags (solid lines) and topics (dotted lines).

*stantially changed* across time. This need is well-reflected in our datasets: About 27% of word types (i.e. unique words) on **CUSTOMNEWS** each month had never occurred in the training period, such as "Brexiteers" and "MeToo". We refer to these as EMERGING NEW WORDS, and argue that these concepts are important because they reflect precisely the dynamic nature of our non-stationary world. Perhaps the most notable recent EMERGING NEW WORDS is "COVID-19", which had zero unigram probability prior to late-2019, and yet constitutes an important use case of the NLP systems today.

Concretely, we define EMERGING NEW WORDS as those that occur frequently on the test set (at least 50 times), but either: (i) were previously unseen on the training set, or (ii) occurred much less frequently on the training set than on the test set, as indicated by an at least 5 times lower unigram probability. This procedure yields a reasonably-sized set of 287 EMERGING NEW WORDS and 87,636 mentions in our 2018-2019 test documents. Many of these words indeed reflect strong temporal dynamics: e.g. "Ardern" (who became the New Zealand PM in late-2017) and "Novichok" (which is what Sergey and Yulia Skripal were poisoned with in 2018). Fig. 3 shows that the TIME-STRATIFIED model performs substantially worse for EMERGING NEW WORDS—an almost 5x worse perplexity (110 vs 22) than the overall one (Figure 2).

**Perplexity of first and second occurrences of EMERGING NEW WORDS.** We now ask: How well can Transformer LMs rapidly adapt to new information and EMERGING NEW WORDS? Concretely, LMs that perform well in our non-stationary world should be able to predict subsequent occurrences

of EMERGING NEW WORDS (e.g. "COVID-19"), which exhibit strong temporal dynamics, much better than the first occurrences of these words, because these words appear frequently on the test set prefix—even though these EMERGING NEW WORDS do not appear as frequently on the training set. In Fig. 3, we show the perplexity obtained by the TIME-STRATIFIED model under two conditions: For the first and second occurrences of EMERGING NEW WORDS in a test document.

Although the model has a high ppl. the first time it generates EMERGING NEW WORDS in the document (ppl. of ∼694.95), it has a much lower ppl. for generating the same words for the second time, but *only if* the first word is available in the Transformer context. In such case, the model can simply copy the same word from the context; this finding reaffirms the strong copying ability of the attention block (Bahdanau et al., 2015; Vinyals et al., 2015). This means that the ability of Transformers to condition on long-range context is *already* a useful feature for temporal generalization, even when we are not explicitly updating the model parameters with new data. However, we observe no such effect when the first occurrence falls *outside* of the Transformer memory (ppl. of >2,700), highlighting the need

| Setup / Occurrence | TIME-STRATIFIED | TIME-STRATIFIED + dynamic eval |
|---|---|---|
| All words | 22.45 | 22.17 |
| All EMERGING NEW WORDS | 109.73 | 66.26 |
| 1st | 694.95 | 357.40 |
| 2nd — 1st in memory | 75.95 | 44.21 |
| 2nd — 1st **NOT** in memory | 2,719.25 | 1,430.34 |

Table 3: Perplexity of TIME-STRATIFIED model on EMERGING NEW WORDS on **WMT**, broken down by whether the word is encountered for the 1st or the 2nd time in the test document, and for the latter, whether the 1st occurrence was in the TXL context. The last column shows results with dynamic evaluation (§6).

to scale Transformers to even longer sequences (Child et al., 2019; Correia et al., 2019; Kitaev et al., 2020; Beltagy et al., 2020, *inter alia*) to improve temporal generalization.

**Importance.**   Our analyses provide a targeted evaluation of temporal generalization in LMs, which enable us to benchmark progress precisely on things that matter the most for temporal generalization (e.g. evaluating LMs on named entities, fast-changing topics, and adaptation speed to EMERGING NEW WORDS, rather than relying on overall ppl. as a sole metric for measuring LM progress).

## 4   The effect of outdated models persists even when increasing model sizes

Recently, increasing model size has led to substantial improvements in perplexity, downstream tasks, and few-shot learning ability (Kaplan et al., 2020; Brown et al., 2020). But can increasing model size also improve temporal generalization? To this end, we train a bigger TIME-STRATIFIED model with 448M parameters—a 60% increase over the previous 287M model and 30% larger than GPT-2$_\text{MEDIUM}$.

Similar to Section 3, we report the respective perplexity increase of the newly trained TIME-STRATIFIED$^{448M}$ model over the CONTROL$^{287M}$ model (solid lines). We reproduce the relative perplexity increase of the smaller TIME-STRATIFIED$^{287M}$ model over the CONTROL$^{287M}$ one (Fig. 2) as the dotted lines.

If increasing the model size was able to delay temporal degradation, we would expect to see the solid lines produced by the bigger models

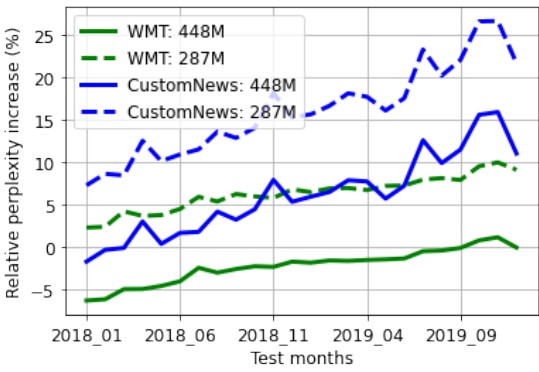

Figure 3: Relative perplexity increase of the TIME-STRATIFIED models with 287M (dotted line) and 448M parameters (solid line), respectively, over the CONTROL model with 287M parameters, for **WMT** and **CUSTOMNEWS** (§4).

to have reduced (i.e., flatter) slopes compared to the dotted lines produced by the smaller models. While larger TIME-STRATIFIED models, as expected, achieve lower absolute perplexities (5.5% improvement), model size has *no significant effect* on the slope of these lines ($p > 0.05$, assessed using a t-test on the slopes found by fitting a linear regression). On both datasets, by the end of the test period (i.e. late-2019), a smaller but more up-to-date CONTROL$^{287M}$ model outperforms a 60% larger but two-year out-of-date TIME-STRATIFIED$^{448M}$ model. Hence, building models that perform

well in this setup requires solutions that more directly tackle the specific challenges we emphasized through our findings so far, and update the model's knowledge with new information.

## 5   Time-stratified question answering

So far we have evaluated the LMs intrinsically, through perplexity, which is important because language modelling is a *foundational* task that affects many NLP applications through language model pretraining. However, we still do not know how this perplexity deterioration affects practical applications of LMs, i.e., how do out-of-date LMs affect different types of downstream tasks?

**Closed-book question answering (QA).** Closed-book QA is a popular testbed for evaluating pretrained LMs that have to compress and encode knowledge found in a big corpus. But given the relative lack of existing time-stamped, news QA datasets that evaluate LMs' ability to answer questions about events that happen outside of their training period in a closed-book fashion, we construct a dataset of synthetic questions the government officials using the following template: "*Who is the [government role] of [country/state] in [month/year]?*" In total, we construct a test set of 438 questions about 22 different government roles from 11 countries (see Appendix C for examples). We pretrain the TXL model (as described in Section 2.3) using the **WMT** dataset up to the years 2012, 2013,. . . , 2019, respectively. We then fine-tune all these models to answer questions about government officials for 2011 to get the model accustomed to the task format, and

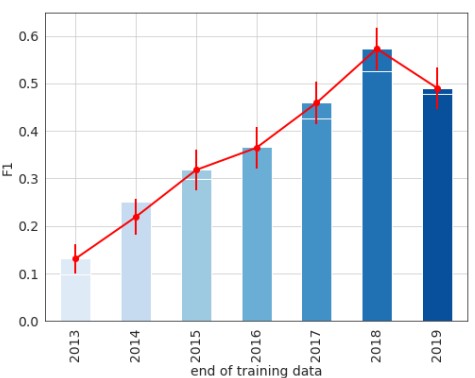

Figure 4: Synthetic questions about political figures: Model performance as we shift the end of the training set chronologically away from the year about which we ask questions (i.e, 2019). The error bars indicate two standard errors of mean.

evaluate on synthetic questions related to the year 2019. Fig. 4 shows the substantial accuracy deterioration as we shift the end of the pretraining data away from 2019, the year for which we ask questions. This finding demonstrates how the fine-tuned LMs' lack of more recent factual knowledge affects their performance on this task. Note that the slight drop in accuracy in 2019 compared to 2018 is due to dataset noise. Anecdotally, we observe that the 2019 model mixes up the names of Russian and American presidents, which often co-occurred in the same context in 2019.

**Reading comprehension.** Nevertheless, we do not expect all downstream tasks to be equally affected by outdated LMs. To illustrate this point, we perform a reading comprehension experiment using NewsQA (Trischler et al., 2017), where the evidence documents are presented together with the questions into the prefix of the model. Hence, the model has all necessary information to answer the questions, and thus outdated LMs will likely present less of a challenge in this type of tasks. We obtain a TIME-STRATIFIED NewsQA by recovering the articles' timestamps.[7]. We test on questions from 2009, for a of total 10000 questions (see Appendix C for question-answer examples). We evaluate how well LMs trained on **CUSTOMNEWS** until the end of 2008 performs in comparison to LMs trained until the end of 2009: Both models perform identically at 0.47 F1. Hence, time-stratified evaluations for reading comprehension, where the answers are extractive and can be copied from the passage, pose less of a challenge for outdated LMs, unlike knowledge-intensive, closed-book QA.

## 6   Keeping models up-to-date: Online learning through dynamic evaluation

One way to mitigate LMs' degradation over time is to continually update the models' knowledge with new information as new documents arrive into the stream. One of the ways to do this is through dynamic evaluation (Mikolov et al., 2010; Graves, 2013; Krause et al., 2019)—a form of online learning that continually updates the parameters of a pretrained model by performing gradient descent on the new data. While most prior work used dynamic evaluation to perform updates within a document, hence adapting to local topical shifts, here we use it to adapt to the temporal dynamics

---

[7]https://cs.nyu.edu/ kcho/DMQA/

that occur within a stream of chronologically ordered documents, hence adapting to temporal trends across documents. Appendix B has more details on dynamic evaluation and our empirical settings.

We plot the results in Fig. 5: Dotted lines reflect the perplexity increase when comparing the CONTROL model to the TIME-STRATIFIED model, i.e., the same graph as in Fig. 1, whereas solid lines reflect the perplexity increase achieved when comparing the same CONTROL model with the TIME-STRATIFIED model augmented with dynamic evaluation (TIME-STRATIFIED$^{dyn}$). In all datasets, dynamic evaluation reduces the speed of the model becoming outdated, as evidenced by the *reduced* upward slope, with a significant effect for **ARXIV** and **WMT** ($p < 0.05$, assessed using a t-test on the slopes found by fitting a linear regression). The improvements are more pronounced for **ARXIV**, where a more granular analysis over weeks reveals that the model needs only about one week worth of data to overtake the CONTROL model. Moreover, we see much larger improvements for predicting EMERGING NEW WORDS, which exhibit strong temporal dynamics (§3.1, see Fig. 3): We observe a 39.62% ppl. reduction from 109.73 to 66.2 for EMERGING NEW WORDS, compared to the overall ppl. reduction (a 1.25% reduction from 22.45 to 22.17 for **WMT**; Fig. 4).

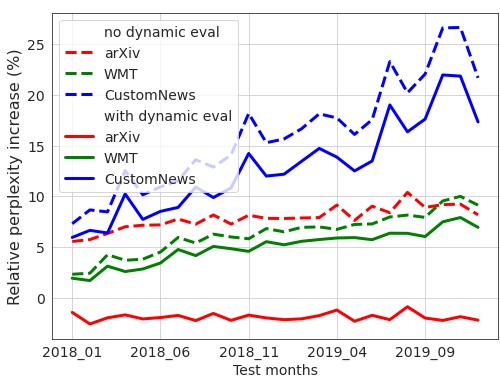

Figure 5: Relative perplexity increase with (solid lines) and without (dotted lines) dynamic evaluation, for the TIME-STRATIFIED model.

When aiming to keep models up-to-date (especially for larger models), lightweight yet effective approaches are preferable because they allow the model to rapidly digest new information with minimal time, computation, and carbon costs. We thus experiment with updating only the embedding layer (i.e., 52M parameters), capturing lexical semantic changes, as well as updating only the bias terms at all layers (i.e., 198K parameters), as recently introduced by Ben-Zaken et al. (2021). Fig. 4 presents the results: In line with the findings of Ben-Zaken et al. (2021), updating only the bias terms performs nearly as well as updating the full model.

| Parameters that get updated | **CUSTOM** | | |
| | **WMT** | **NEWS** | **ARXIV** |
|---|---|---|---|
| All parameters | 22.17 | **20.72** | **20.98** |
| Bias-only | **22.16** | 20.96 | 21.24 |
| Embeddings-only | 22.32 | 21.21 | 22.27 |
| no dynamic eval. | 22.45 | 21.33 | 23.07 |

Table 4: Perplexity of TIME-STRATIFIED model when updating some of the parameters.

**Beyond dynamic evaluation** We remark that dynamic evaluation alone, while effective, does not fully solve temporal generalization, as evidenced by the prevailing (albeit gentler) upward slopes on **WMT** and **CUSTOMNEWS**. We expect that larger gains can be achieved by fully embracing continual learning in LMs—striking a balance between quickly adapting to new data and achieving positive forward transfer (Lopez-Paz and Ranzato, 2017), while avoiding catastrophic forgetting (Mccloskey and Cohen, 1989; Kirkpatrick et al., 2017). Indeed, as we show in Figure 9, while dynamic evaluation is able to improve generalization to future data, it causes catastrophic forgetting of the past data. Furthermore, recent semi-parametric models (Guu et al., 2020; Lewis et al., 2020b; Karpukhin et al., 2020; Khandelwal et al., 2020; Yogatama et al., 2021) lend themselves well to continual learning, where new knowledge can be stored in an external memory, which can be updated without retraining the whole model. A related approach is to disentangle the acquisition of up-to-date knowledge from the language learning itself, and enable direct editing of that knowledge within the model (Sinitsin et al., 2020; Zhu et al., 2020; De Cao et al., 2021).

## 7 Related Work

**Concept drift.** Detecting changes in data streams, also known as concept drift, has a long history in machine learning (Widmer and Kubat, 1996; Kifer et al., 2004; Baena-García et al., 2006; Dries and Rückert, 2009; Lu et al., 2019). In NLP, most recent work in this area models lexical change by

training word embeddings (Hamilton et al., 2016; Szymanski, 2017; Yin et al., 2018) and deep neural networks (Rosenfeld and Erk, 2018; Bjerva et al., 2019) on data of different time spans.

**Out-of-Distribution (OoD) generalization.** OoD generalization is well-studied in NLP (Blitzer et al., 2006; Daumé III, 2007; Axelrod et al., 2011), and has recently been addressed for neural LMs and transfer learning (Fried et al., 2019; Oren et al., 2019; Gururangan et al., 2020), where pretrained LMs lead to substantial improvements and increased robustness (Hendrycks et al., 2020). While most prior work has focused on distributional shifts in terms of topic and domain (Kruszewski et al., 2020), distributional shifts in terms of *time* also constitute an important and realistic challenge that NLP systems of today (including those based on large LM pretraining) must be able to perform well at.

**Continual learning & streaming LMs.** Our work is closely related to continual and lifelong learning, which aim to design models that continually accumulate new knowledge without forgetting relevant information about the past (Mccloskey and Cohen, 1989; Thrun and Mitchell, 1995; French, 1999; Mitchell et al., 2015; Rusu et al., 2016; Kirkpatrick et al., 2017; Al-Shedivat et al., 2018; Hadsell et al., 2020). The distribution of words and context in natural language changes rapidly with time, and hence constitutes an important test bed for developing continual learning systems. More specific to the LM literature, prior work proposed ways of designing LMs that efficiently adapt their knowledge to continuous streams of new information (Jelinek et al., 1991; Wang et al., 2008; Goyal et al., 2009; Osborne et al., 2014; Yogatama et al., 2014, *inter alia*)—often known as *streaming LMs*, albeit mostly in the context of $n$-gram LMs. While we show that Transformer LMs achieve much better perplexity than previous $n$-gram models (the Transformer LM ppl. in §3 are substantially better than those in prior streaming $n$-gram LM literature), we show that Transformer LMs similarly suffer from the temporal degradation problem. Given the different nature of our LMs today (i.e. deep neural models rather than $n$-gram LMs), we argue that now is the right time to make progress on this open research question, with notable progress in other NLP tasks (d'Autume et al., 2019; Sun et al., 2020).

**Temporal splits in NLP.** Prior work has used temporal splits (i.e. training on text from the past and evaluating on future text) for NLP tasks like machine translation (Levenberg et al., 2010), sentiment analysis (Lukes and Søgaard, 2018), named entity recognition (Fromreide et al., 2014; Rijhwani and Preotiuc-Pietro, 2020), and others (Dunn et al., 2017; Bjerva et al., 2020; Søgaard et al., 2021). Nevertheless, the vast majority of NLP benchmarks today still do not perform temporal splits, and hence do not measure how well models can generalize to future data. Furthermore, this work has two key distinctions from prior work. First, we focus on language modelling—a *foundational* task that is used for many NLP systems today through LM pretraining—and propose a benchmark to measure progress in this direction. Second, we go one step further than prior work and perform a thorough analysis to pinpoint *what kinds* of predictions the model is struggling with. Such analysis can then be used to better measure progress in dynamic language modelling, where improvements are not always easily discernible from overall ppl. alone (e.g. performance on EMERGING NEW WORDS; §3.1).

## 8 Conclusion

We evaluated the extent to which our current language models can generalize well in the realistic setup of predicting future utterances outside their training period. We found that current practices that train and evaluate models on data from overlapping time period *overestimate* model generalization to future utterances, and that Transformer LMs become increasingly outdated with time. We found that increasing model size alone—a key driver behind recent LM success—is not a solution for this problem, and that this degradation affects downstream tasks that require up-to-date factual knowledge.

**Generality to other domains.** The importance of temporal generalization extends beyond language modelling and NLP. Many other commercial machine learning systems like speech processing and video understanding also involve non-stationary data, and are similarly trained on data that were collected in the past, but deployed on new data from the future. Since many of these tasks also use Transformers (Girdhar et al., 2019; Gulati et al., 2020, *inter alia*), we expect our findings to generalize to these domains, although a more complete investigation to this end is left to future work.

**Limitations** While we explored how the LM performance degradation with time affects two types of question-answering tasks, a broader variety of tasks is needed to obtain a more holistic picture on how temporal generalization manifests in downstream tasks. An open research question is thus how we can create and maintain benchmarks that are not static (Nie et al., 2020; Potts et al., 2020) and further promote research on continual and life-long learning.

**Is this all obvious?** Our findings should not come us a surprise: That the world around us changes with time—and thus *what* and *how* we talk about it also evolve accordingly—is hardly controversial. But for the most part, these temporal dynamics are *still not* currently reflected in the way that we train

and evaluate our neural LMs. Our aim here is to highlight how such static evaluations overestimate models' performance, especially around predictions that relate to factual knowledge, which constitute an important use case of NLP systems today. With the compilation of ever-larger web crawls for LM pretraining (Gao et al., 2021), now is the right time to rethink how our splits are constructed (Søgaard et al., 2021), construct temporal splits that evaluate models on their ability to generalize to future data, and include timestamp information in both pretraining datasets and downstream tasks to facilitate this kind of more realistic LM evaluation. This strategy will not only allow us to assess models' performance on a realistic type of out-of-sample data, but also circumvent test data contamination affecting both LM and downstream task evaluations more broadly, e.g., in widely used QA datasets like Natural Questions (Kwiatkowski et al., 2019) and TriviaQA (Joshi et al., 2017a), which have been shown to contain alarmingly high proportions of overlapping training and test data (Lewis et al., 2020c). Our dynamic, streaming LM benchmarks—alongside the evaluation metrics that evaluate LMs on things that matter the most for temporal generalization (e.g. named entities, EMERGING NEW WORDS)—will be released to encourage more research in this area, and reward the development of **adaptive** language models that can remain up-to-date with respect to our non-stationary world.

## 9 Broader Societal Impact Discussion

Lastly, we remark on two aspects of the broader societal impact pertaining to the importance of continually-updating language models. First, we argue that having NLP models—the vast majority of which are built on top of pretrained language models—that remain up-to-date with our current social trends and public perception is relevant for mitigating potential harms and biases caused by NLP models. For instance, recently there has been renewed public support and interest for social justice movements in 2020, such as the #BlackLivesMatter movement (Cohn and Quealy, 2020). Hence, without explicit mechanisms to update the models' knowledge, language models that were trained before this time period can potentially miss out on shifting language used to describe such movements—where such movements are now more widely supported by the general public—and potentially produce outdated, biased language that is no longer frequently used at present. On the other hand, we should also be careful not to let the model update its knowledge with material that can add or amplify to the bias and prejudice of the model (Chang et al., 2019; Bender et al., 2021).

Second, our findings highlight the risk of the "brute-force" approach of keeping models up-to-date by periodically retraining the model from scratch, for instance by combining the old and new data. Given the increasing size of NLP models, training large models from scratch each time incurs increasingly more expensive computational and environmental costs (Strubell et al., 2019; Patterson et al., 2021). Hence, our findings emphasise the need for more efficient and lightweight approaches of keeping models up-to-date, whilst mitigating catastrophic forgetting at the same time. Our work provides a benchmark to measure progress in this space, and we strongly encourage future work that uses our benchmark to also report the computational costs of their approach for keeping language models up-to-date. Lastly, we remark that the ethical considerations and risks of working with large language models also apply to our work (Bender et al., 2021).

### Acknowledgments and Disclosure of Funding

We thank Paul Michel, Laura Rimell, and Chris Dyer for useful feedback throughout the different stages of this project. We would also like to thank Katie Millican, Sebastian Borgeaud, Trevor Cai, Roman Ring, Jack Rae, and Geoffrey Irving for their initial work on the codebase. The authors received no specific funding for this work.

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
