# A How General Are These Findings?

## A.1 The effect of outdated models persists beyond the 2018/2019 test period.

We test whether the temporal degradation trends we observe in §3 are *not* an artifact of some particularity of the chosen test period (i.e., $Yr1 = 2018$ and $Yr2 = 2019$). We design new test sets by shifting $Yr1$ and $Yr2$ in increments of one year towards the past, for a total of five such test sets. Following §2.2, we derive different TIME-STRATIFIED$^{Yr1,Yr2}$ and CONTROL$^{Yr1,Yr2}$ training and validation splits.

Note that each TIME-STRATIFIED$^{Yr1,Yr2}$ and CONTROL$^{Yr1,Yr2}$ setups are: (i) Trained on the same training data sizes, and (ii) evaluated on the same test set covering $Yr1$ and $Yr2$. Fig. 6 shows similar temporal degradation across all testing years.

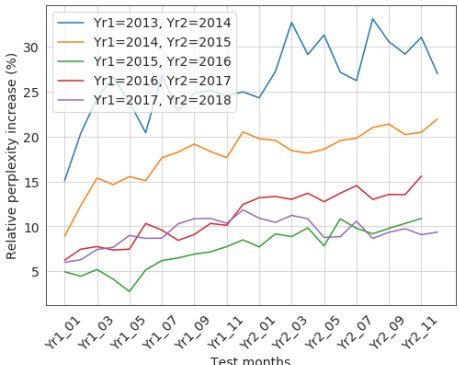

Figure 6: Relative perplexity increase of TIME-STRATIFIED$^{Yr1,Yr2}$ over CONTROL$^{Yr1,Yr2}$ models.

## A.2 The effect of outdated models persists beyond the two-year gap.

For this experiment, we keep the same 2018-2019 test set introduced in §2.2, and train models with training data from different time periods with increasingly larger gaps from the 2018-2019 evaluation period, controlling so that all training data sizes are identical across different years. More concretely, the most up-to-date model covers the same time period as the original TIME-STRATIFIED model, and we "push" the training period back with 6-month increments, up to September 2012, for a total of 11 training sets—each of the same size—used to train 11 models. Fig. 7 shows that the perplexity deterioration continues to grow in response to larger gaps between the training and test periods.

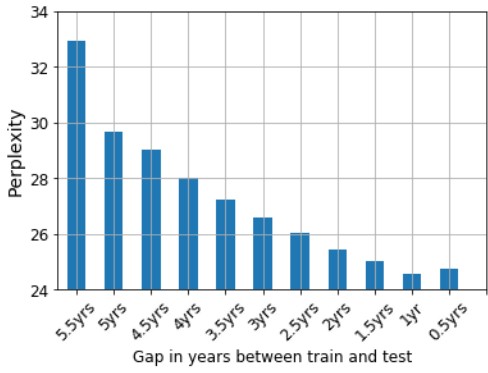

Figure 7: Perplexity of models trained with data from different time periods, with increasingly larger gaps from the 2018-2019 test set period.

### A.3 The effect of outdated models persists beyond English: A German study.

We test whether the temporal degradation trend is a generalizable pattern that holds across languages. We use the German subset of **WMT**, apply the same pre-processing steps as §2.1, follow the same experimental setup as §2.2, and train two Transformer-XL models on TIME-STRATIFIED$^{de}$ and CONTROL$^{de}$ setups, achieving 30.87 and 26.79 respective test perplexities. These perplexities are indeed higher than the ones in Table 2—a consistent pattern with prior findings on the difficulty of modelling German (Mielke et al., 2019). Nevertheless, we still see the exact same pattern where the stale TIME-STRATIFIED$^{de}$ model performs worse than the CONTROL$^{de}$ one (a substantial 15.23% relative increase). Moreover, similar to the English experiment, the model degrades more as the gap between the training and test period increases—an effect particularly pronounced for proper nouns and for words that are broken down by the TIME-STRATIFIED$^{de}$ tokenizer into more tokens.

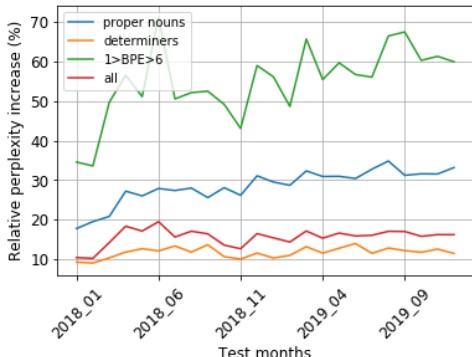

Figure 8: For the experiments on German, the relative increase of perplexity of the TIME-STRATIFIED$^{de}$ model over its CONTROL$^{de}$ counterpart.

## B Dynamic evaluation

Here we more formally describe dynamic evaluation, which we apply to the TIME-STRATIFIED model, and outline some of the hyper-parameter choices used for our dynamic evaluation experiments (§6). Let $\{D^{(1)}, D^{(2)}, \cdots, D^{(N)}\}$ be a collection of $N$ chronologically-ordered test documents, where $D^{(t-1)}$ was published before $D^{(t)}$, and $D^{(1)}$ was our first test document in the 2018-2019 evaluation period (§2.1). Each test document $D^{(t)}$ consists of $M = |D^{(t)}|$ tokens $\mathbf{x}^{(t)} = x_1^{(t)}, x_2^{(t)}, \cdots, x_M^{(t)}$. Furthermore, let $\boldsymbol{\theta}_1$ be the set of Transformer-XL model parameters (§2.3) *after* training on documents from the pre-2018 training period (TIME-STRATIFIED setup; §2.1), and *before* any dynamic evaluation is applied.

The loss of the Transformer-XL model with respect to a test document $D^{(t)}$ is computed as follows:

$$\ell(D^{(t)}; \boldsymbol{\theta}_t) = \log p_{\boldsymbol{\theta}_t}(\mathbf{x}^{(t)}) = \log \left( \prod_{i=1}^{M} p_{\boldsymbol{\theta}_t}(x_i^{(t)} \,|\, \mathbf{x}_{<i}^{(t)}) \right) = \sum_{i=1}^{M} \log p_{\boldsymbol{\theta}_t}(x_i^{(t)} \,|\, \mathbf{x}_{<i}^{(t)}), \qquad (1)$$

where $\mathbf{x}_{<i}^{(t)}$ denotes tokens $x_1^{(t)}, x_2^{(t)}, \cdots, x_{i-1}^{(t)}$ in the test document $D^{(t)}$ that precede $x_i^{(t)}$.

In dynamic evaluation (Mikolov et al., 2010; Graves, 2013; Krause et al., 2018, 2019), we dynamically update the Transformer-XL model parameters using gradient descent, based on the knowledge contained in the test documents that had been seen so far. More formally,

$$\boldsymbol{\theta}_{t+1} \leftarrow \boldsymbol{\theta}_t - \alpha \, \nabla_{\boldsymbol{\theta}_t} \, \ell(D^{(t)}; \boldsymbol{\theta}_t), \qquad (2)$$

where $\alpha$ denotes the dynamic evaluation learning rate, and $\nabla_{\boldsymbol{\theta}_t} \, \ell(D^{(t)}; \boldsymbol{\theta}_t)$ denotes the gradient of the model parameters with respect to the model's loss for the current document $\ell(D^{(t)}; \boldsymbol{\theta}_t)$.

This procedure means that the model parameters $\boldsymbol{\theta}_t$, which we use to evaluate the model on the current test document $D^{(t)}$, *already encodes* knowledge from previous test documents

$D^{(1)}, D^{(2)}, \cdots, D^{(t-1)}$, in addition to the knowledge learnt from the training set. This in turn enables the model to learn about new information that emerges or becomes more salient during the evaluation period (e.g. "COVID-19" in late-2019), which is then stored in the model parameters, and reuse such information for better prediction of subsequent test documents. In practice, our implementation of dynamic evaluation differs from Eq. 2 in two ways: (i) We perform $K$ steps of gradient descent for each document, rather than only one step, where $K$ is tuned on the validation set; and (ii) we perform the gradient updates for a batch of contiguous tokens (e.g. 512), which means that documents that are longer than the batch size will have more than one parameter update.

**Contrast with non-dynamic evaluation.** When dynamic evaluation is not applied, $\boldsymbol{\theta}_t = \boldsymbol{\theta}_{t-1} = \boldsymbol{\theta}_1$. This means that the same model parameters $\boldsymbol{\theta}_1$ (i.e. model parameters after training on the training documents—*without* updating the models' knowledge on the observed test documents) are used to predict all test documents, risking the model becoming outdated in-between retraining cycles.

**Dynamic evaluation hyper-parameters.** We use the following learning rates (**WMT**: 5e-5, **CUS-TOMNEWS**:5e-4, **ARXIV**: 1e-3), which are tuned on the validation set spanning three months, whereas the test set spans two years. We leave the question of choosing a learning rate with an optimal trade-off between adaptation speed and stability of updates without *a priori* knowledge of the evaluation period to future work.

### B.1 Dynamic Evaluation and Catastrophic Forgetting

We design an experiment to assess whether updating a model on present data using dynamic evaluation leads to catastrophic forgetting of the past data. To assess this, we report the performance of the two models, i.e., the one trained until 2017 and the one updated up to 2019, on a test set derived from the initial training data of the model covering the years up to the year from which we started performing dynamic evaluation (i.e., 2007-2017). In addition, we also report the results on the 2018-2019 test set which were presented in Section 6.

Figure 9 presents the results for **WMT** and**ARXIV**. For both datasets we observe that as we move towards the past, the perplexity of the model updated with dynamic evaluation increases. As such, while the updated model outperforms the outdated model for the recent 2018 and 2019 years, the same model performs increasingly worse on the past years, as indicated by the gentle upward slope from 2017 and onwards.

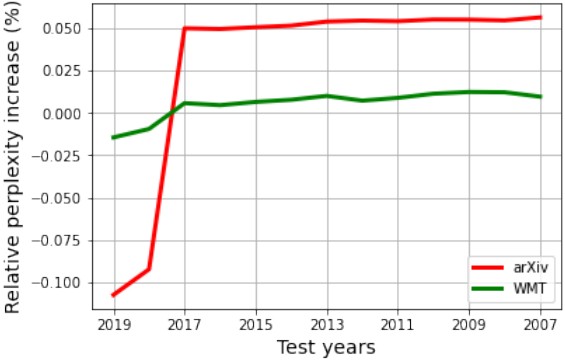

Figure 9: Catastrophic forgetting as measures in terms of relative perplexity increase when comparing the models updated with dynamic evaluation against the models that have been trained with data up to 2017. The x-axis presents the years in a reverse chronological order.

## C Example Question-Answer Pairs

### C.1 Examples of closed-book QA on synthetic questions on government officials

**Question:** Who was the governor in Texas on 5 September 2019? **Answer:** Greg Abbott

**Question:** Who was the prime minister in Canada on 8 June 2019? **Answer:** Justin Trudeau

**Question:** Who was the president in Portugal on 30 May 2019? **Answer:** Marcelo Rebelo de Sousa

## C.2 Examples of reading comprehension on NewsQA

**Document:** England international footballer Steven Gerrard was found not guilty of affray by a court in his home city on Friday. England international Steven Gerrard was cleared by a court in Liverpool of affray. The jury at Liverpool Crown Court took a little over an hour to clear Gerrard of charges relating to a fracas in a nightclub bar in the north-western of England city on December 29 of last year. They accepted the Liverpool captainś version that he acted in self defense in punching businessman Marcus McGhee. The 29-year-old was the only one of the seven defendants in the case to be cleared after an incident which was described by judge Henry Globe as an "explosion of violence." Gerrard spoke of his relief outside the court. "Can I just say how pleased I am with todayś verdict," he said. "Iḿ glad to put this case behind me and I am really looking forward to the season ahead and concentrating on my football now. "I would just like to say a big thank you to my legal team and to my friends and family and everyone at Liverpool football club for supporting me." His comments were met with a round of applause from a large group of fans of the Premier League club who had gathered outside the court, before he was ushered away. Gerrard was celebrating in the Lounge Inn in Southport, a suburb of Liverpool, after scoring twice his teamś 5-1 win at Newcastle which took them to the top of the Premier League. Video footage, which was available to the court, showed.

**Question:** Who was cleared by a Liverpool court? **Answer:** Steven Gerrard

**Document:** CNN affiliates report on where job seekers are finding work across the country and how those looking for employment are coping with the situation. A census employee poses with the new handheld device field workers will use for the 2010 count. (CNN) – The nation will take roll call in 2010 and the federal government is giving the states money to hire thousands of census workers. Officials in Colorado say they may hire as many as 8,000 workers for positions that last between 10 weeks and one year. Cathy Illian says the bureau has already hired 800 people in the Denver area. The organization will also post open positions in early April. Some jobs pay as much as $28.75 an hour. Read the story on KMGH. In Idaho, Dave Mulvihill, manager of the stateś census bureau, said the organization will hire 1,200 workers. He has plenty of job searchers to choose from. "WeV́e had applications from approximately 7,300 people across the state," he told CNN affiliate KIVI. Read the full report on census jobs. The office is holding off on taking any more applications until fall. The Alabama census bureau is preparing to hire between 1,000 and 1,500 workers. "We need workers so we can get good addresses [to] send the questionnaires out so we can get a good response," state census bureau official Darryl Lee told TV Alabama in Birmingham. Census officials point out that an accurate count of U.S. citizens helps the government figure out how much funding to give each state for federally sponsored programs. Read the ABC 33/40 story Northeast: Rhode Island strip club.

**Question:** Census bureaus are hiring people from where? **Answer:** Denver area