# OpenReview forum: "Mind the Gap: Assessing Temporal Generalization in Neural Language Models"
_NeurIPS.cc/2021/Conference — NeurIPS 2021 Spotlight_

### Official Review · Reviewer_1qBQ · 2021-07-12

**Rating:** 6
**Confidence:** 4

**Summary:**

This paper evaluates language models on time-stratified datasets, i.e., they split datasets so that the test set is newer in time than the training set. It shows LMs are worse in terms of perplexity and closed-book question answering ability on the test data, some times by surprisingly large amounts. They analyze where the largest source of errors occurs (proper nouns, named entities, etc.) and also show that online finetuning at test time can mitigate but not solve these temporal generalization issues.

**Limitations And Societal Impact:**

Yes.

**Main Review:**

Strengths:
* Well-written, good motivation, and an understudied problem in the NLP community.
* Simple and convincing experiments. While some may critique this paper as being "obvious", its a study that has seen little exploration and is definitely worth doing.
* I like that they explored ways to mitigate the problems with dynamic evaluation, even including lightweight updates, which is the most sensible baseline strategy for adapting models (although see weaknesses section)

Weaknesses:
* I'm actually surprised by how relatively ineffective dynamic evaluation is, especially for the news data. It'd be great to know why. One theory, for example, is that catastrophic forgetting is occurring and the LM is being adapted to the news domain but becoming worse in overall performance. A simple test for this would be to run the model on held-out data that is not temporally shifted and see how much perplexity changes before/after dynamic evaluation.



**Time Spent Reviewing:**

2

---

> ### Author Response · Authors · 2021-08-10
> **Author response**
>
> We thank the reviewer for their insightful comments and valuable suggestions. We are delighted that the reviewer finds the paper to be well-written, well-motivated, and convincing in terms of empirical findings.
>
> The suggestion on quantifying whether, and to what extent, catastrophic forgetting is happening after dynamic evaluation is a great one. To this end, we have conducted a preliminary experiment, where we evaluate the model checkpoints before and after applying dynamic evaluation. We evaluate both models on a subset of training and validation documents from arXiv and WMT, where these documents come from the past 10 years. Indeed, we find that the dynamic evaluation model outperforms the non-dynamic evaluation model towards the end of this 10-year period (e.g. year 2018), which is consistent with the findings in the paper. However, the dynamic evaluation model performance starts degrading (relative to the baseline model without dynamic evaluation) as we move towards the beginning of the 10-year period (e.g. year 2008). This finding indicates that catastrophic forgetting is indeed an issue with dynamic evaluation; we further find a more pronounced catastrophic forgetting effect for the arXiv dataset. We commit to conducting a more thorough experiment on this topic for subsequent versions of this paper.
>
> Above and beyond the risk of catastrophic forgetting, we outline three further limitations of dynamic evaluation in its current form, which might not render them particularly effective for our dataset. In future work, we plan to address and improve upon these limitations.
> * First, most current state of the art language models succeed when there is a large amount of data to learn from (e.g. GPT-3, BERT, etc.), such as web-crawled English text. Dynamic evaluation, however, is performed on the test data that has been observed so far, but the amount of test data is often much smaller than the set of pretraining data used to originally train the language model. Performing gradient descent updates on this relatively small test data can thus potentially introduce some instability. In our experience, we have to tune the learning rates carefully to avoid large performance swings due to instability when performing dynamic evaluation. Since we cannot assume access to the test set beforehand, we can only tune this dynamic evaluation learning rate on the validation set, which spans only 3 months in our case, while the test set can span multiple years. The small size of the test set, combined with the large size of the model, further introduces the risk of overfitting to the particular test period for which we are performing gradient updates.
> * Second, dynamic evaluation requires updating the whole parameters of the model, although we found that updating only the embedding layer performs only slightly worse (Table 4). Nevertheless, not all aspects of natural language change rapidly with time. The syntax of the English language, for instance, has changed slowly over the years, while the lexicon of the English language has expanded rapidly over the same period (e.g. new words and concept like “COVID-19” and “#BlackLivesMatter”). We conjecture that dynamic evaluation would work better and be more data-efficient in models that factorise its knowledge over different parts of language (e.g. separate modules for syntax and lexicon), where only the relevant subparts of the model need to be updated with new knowledge.
> * Third, we conjecture that other machine learning techniques—like meta-learning/learning-to-learn—can potentially improve the performance of dynamic evaluation, since some of these techniques aim to make the model learn as efficiently as possible from a few steps of gradient descent. Combining meta-learning with dynamic evaluation thus constitutes a promising avenue for designing language models that can remain up-to-date with respect to our non-stationary world, which we plan to explore in future work.

---

### Official Review · Reviewer_bwt6 · 2021-07-15

**Rating:** 8
**Confidence:** 4

**Summary:**

This paper studies temporal generalization of language models -- how well they do on text data "from the future" (ie after they are trained). It performs experiments in the domains of arxiv abstracts and news domains, where the date of each document is given.

The paper trains language models on data up to and including December 2018, and evaluates perplexity on the months afterwards -- perplexity goes up as models have to extrapolate more, much more than if documents were split iid. The paper finds similar reslults when the task is changed to QA, or when a larger model is pretrained.

The paper introduces analysis for what new linguistic patterns cause perplexity to increase (e.g. new words or words that change in context.)

The paper tries one simple way of addressing this -- dynamic evaluation -- but also argues that it's not enough.

**Ethical Concerns:**

None that come to mind to this reviewer.

**Limitations And Societal Impact:**

This paper discussed "limitations" but it did not discuss possible negative societal impact of the work -- the paper could be improved by adding such a section. Even though this benchmark is not likely to cause harm, it might be good to think about what might result from future work in this area (on both modeling and evaluation side, with respect to the field and society). You could also discuss the relevance of evaluation datasets like this one in terms of reducing harms of NLP (e.g. bias being amplified? or historical prejudices being baked in?).

**Main Review:**

To this reviewer, this paper seems strong. Generalizing to the future is a challenging task and this paper helps shed some insights on where today's models struggle. The dynamic evaluation baseline should also help ground future work in this direction.

Releasing these benchmarks should help future work in this area, particularly if the underlying metrics (like performance on "emerging new words") can be accessed by submitters.

To this reviewer, there aren't any significant issues with the paper that make me want to reject it. Though the results might not be surprising (discussed well in L439) this paper defines the task in a good way for other work to (possibly?) take a stab at it.

**Time Spent Reviewing:**

1

---

> ### Author Response · Authors · 2021-08-10
> **Author response**
>
> We thank the reviewer for their detailed summary of our work and valuable suggestions. We are pleased that the reviewer finds the paper to be strong; we are also glad that the reviewer recognises the potential impact of our work in grounding and encouraging more research in models that can generalise to our dynamic and non-stationary world.
>
> As suggested by the reviewer, we commit to releasing not only the benchmark, but also the more targeted metrics that we used to specifically evaluate temporal generalisation ability and adaptation speed to new words and concepts, for instance by releasing the set of “emerging new words”.
>
> We also thank the reviewer for the great suggestion to expand on the potential societal impact of our work. In particular, there are two things that we believe are relevant to the societal impact of our work; we briefly include these points below and commit to expanding them in more detail in subsequent versions of our paper.
> * We argue that having NLP models (the vast majority of which are currently built on top of pretrained language models) that can remain up-to-date with our current social trends and public perception is relevant for mitigating potential harms and biases caused by NLP models. For instance, there has been renewed public interest and increased support for social justice movements in 2020, such as the #BlackLivesMatter movement (Bender et al., 2021). Hence, without explicit mechanisms to update models’ knowledge, models that were trained before that time period can potentially miss out on shifting language used to describe such movements (where such movements are now more widely supported by the general public), and potentially produce outdated, biased language that are no longer frequently used at present. On the flip side, we should also be careful not to let the model update its knowledge with materials that can add or amplify the bias and prejudice of the model; how we can best do so remains an important open research question. Furthermore, our dataset also lends itself well to analysing how biases in language models change over time—an important yet relatively under-explored open research question in the field.
> * Furthermore, our paper also highlights the risk of the “brute-force” approach of keeping models up-to-date by periodically retraining the model from scratch, by combining the new and old data. Given the increasing size of NLP models, training large models from scratch each time has an expensive computational and environmental costs, which emphasises the need for more efficient and lightweight approaches of keeping models up-to-date, whilst mitigating catastrophic forgetting at the same time. Our work provides a benchmark to measure progress in this space, and we strongly encourage future work that uses our benchmark to also report the computational costs of their approach.
>
> In subsequent versions of the paper, we commit to including these points concerning the societal impact of our work, and highlight the relevant connections to prior work.

---

> > ### Comment · Reviewer_bwt6 · 2021-08-27
> > **thanks!**
> >
> > thanks for the response! Expanding on societal impact should definitely improve this work.

---

### Official Review · Reviewer_PJPM · 2021-07-17

**Rating:** 8
**Confidence:** 4

**Summary:**

Motivated by the fact that in practice models are applied to text generated later than their training data, this paper shows that splitting the datasets in the common time-agnostic way results in overestimating their performance compared to using time-stratified datasets. This work also conducts in-depth analyses of the performance degradation as the time span from a training set to a (later) test set increases and draws several conclusions: 1) proper nouns, numbers, words related to politics/sports, emerging new words are affected the most; 2) increasing model size improves overall performance but does not solve this performance degradation issue; 3) this phenomenon affects some end-user applications. To mitigate this issue, this work proposes to use dynamic evaluation which updates the model in an online fashion. Based on these findings, this work recommends the wide adoption of time-stratified dataset splits to encourage more realistic evaluation and the development of more adaptive language models.

**Limitations And Societal Impact:**

Yes

**Main Review:**

Strengths:
1. This work identifies a practical yet often overlooked issue of existing text generation evaluation protocols. The wide adoption of time-stratified data would lead to evaluations more consistent with the real-world performance of models deployed.
2. The in-depth analyses of this work find where existing models need improvements when evaluated on time-stratified datasets. This points to future research directions such as incorporating longer contexts to improve the second appearance of emerging new words, or to consider alternative continuous learning approaches rather than simply increasing model sizes to do well under the proposed evaluation protocol.
3. The release of streaming arxiv and news datasets would bolster research on the online adaptation of neural models.

Weaknesses:
1. While it's good to quantify the performance of models on time-stratified datasets, many conclusions are obvious, such as the synthetic experiments on question answering by shifting the end year of training data, as well as the experiments on self-containing reading comprehension.

Typos:
1. L149: Fig. 2 -> you meant Table 2?
2. L234: Fig. 3 -> Table 3
3. L245: Fig -> Table

While the conclusions in this paper might appear obvious, the analyses of this paper are thorough and insightful and the released datasets appear to be useful. More importantly, the recommendation of this paper is relevant to the broad community beyond researchers in text generation. Therefore, I recommend the acceptance of this paper.

==== Post Rebuttal ====
I've read all reviews and the authors' responses. Seems that there's no significant issue that I overlooked in my initial review, so I'm keeping my score and still recommending the acceptance of this paper.

**Time Spent Reviewing:**

2

---

> ### Author Response · Authors · 2021-08-10
> **Author response**
>
> We thank the reviewer for their detailed comments and insights. We highly appreciate the reviewer’s remark that our paper contains thorough and insightful analyses, and that the findings and the proposed dataset from our paper will be useful and of interest to the broader community.
>
> We acknowledge that one shortcoming of our work is that the experimental results on downstream QA tasks are not particularly surprising, as noted by the reviewer. Nevertheless, the findings of our question-answering experiments clearly highlight the need for more and better benchmarks that can assess continual learning ability in important downstream NLP tasks like question answering. Indeed, our paper has only scratched the surface of this important research direction, and we hope that our work will encourage further research by the broader community in this area.
>
> We also thank the reviewer for the typos and presentation suggestions; we will be sure to fix them in subsequent iterations of the paper.

---

> > ### Comment · Reviewer_PJPM · 2021-08-31
> > **Keeping My Score (8)**
> >
> > Thanks for the response! I don't see any significant issue after reading other reviews and your responses. Therefore, I'm keeping my score.

---

### Official Review · Reviewer_vYE6 · 2021-07-19

**Rating:** 7
**Confidence:** 4

**Summary:**

The paper studies the problem of temporal generalization in pre-trained language models, that is, the ability of the models to generalize well to future data beyond their training period. The paper argues that temporal generalization is necessary for pre-trained language models as they increasingly form part of dynamic and non-stationary environments that may require updated knowledge to perform better. To benchmark temporal generalization of the current models, the paper introduces three language modeling datasets - arXiv, WMT, CustomNews with timestamp information. On experimenting with the existing model, Transformer-XL, on the time-stratified setup, the paper finds that the performance deteriorates as the evaluation data moves further away from the training period. An in-depth analysis of the model predictions leads to the intuitive conclusion that the model performance suffers from the novel and emerging words/ concepts present in the future data. Next, the paper considers time-stratified closed-book QA tasks to evaluate the temporal performance of language models on downstream tasks. This paper convincingly shows that the performance on QA tasks deteriorates as the pre-trained language model becomes increasingly outdated with time, thus motivating the temporal generalization problem. Naively increasing the capacity of the model (from Transformer-XL) does not solve the problem. Therefore, the paper builds on the existing dynamic evaluation methods and proposes continuous updating of the language model with the incoming stream of the data. Such a method adapts the language model to temporal trends from incoming data, thus reducing the speed of the model from becoming outdated. Although dynamic evaluation results in improved performance, the paper shows that temporal generalization in pre-trained language models is far from solved and still poses a challenging problem for the community.

**Ethical Concerns:**

No!

**Limitations And Societal Impact:**

Yes, the paper discusses the limitations of their work.

**Main Review:**

The paper investigates temporal generalization in language model pretraining. Although the evaluation task and benchmarks are novel, some recent works are studying this kind of generalization on tasks like sentiment analysis, machine translation, and named entity recognition.  This paper seems an obvious extension to these works with evaluations on a generic language modeling task. Moreover, the considered online learning approach is a straightforward application of the existing dynamic evaluation method. Despite these concerns, when it comes to the novel experimental setup on language model pretraining, the paper is thorough and coherent with well-defined objectives and answers. This paper examines an important problem of temporal generalization that would be prevalent in the coming days. Apart from practical considerations of developing methods to avoid model retraining, considering temporal splits also helps prevent data contamination. The paper promises to release the introduced benchmarks which would benefit the next generation of models and learning algorithms for the temporal generalization problem.


Clarification question: For closed-book QA, in Fig 4 the performance of the model when pre-trained on a dataset up to 2019 drops when evaluated on a test dataset from 2019. The paper attributes the performance drop to confusion around the names of presidents. However, the pre-trained model is fine-tuned on the QA dataset from 2011. Does the model undergo forgetting when fine-tuned on factual knowledge from 2011?


Typos: Lines 234, 245, 361 should be Table 3 instead of Fig 3. Line 368 should be Table 4 instead of Fig 4.

**Time Spent Reviewing:**

4

---

> ### Author Response · Authors · 2021-08-10
> **Author response**
>
> We thank the reviewer for their detailed comments and summary of our work. We are glad that the reviewer finds the paper to be thorough, coherent, and well-motivated in terms of its objectives and underlying research question.
>
> We are happy to clarify the question regarding the closed-book QA experiment, about the performance drop when the base language model is trained with data from up to 2019, relative to the slightly better performance when the base language model is trained only up to 2018 data.
> * While this result is indeed rather counter-intuitive, we remark that the error bars (as shown in red in Figure 4) between the two results (i.e. when training the base LM up to 2018 or 2019, and then fine-tuning them both on the 2011 QA dataset to get the models accustomed to the task format) have some overlap. These error bars indicate two standard deviations of the mean, which we obtain by performing the same experiment multiple times from different random seeds. Since the error bars overlap, one possibility is that the difference between the 2018 and 2019 performance can be attributed to noise.
> * The reviewer suggests that the performance drop when using data up to 2019 can potentially be attributed to catastrophic forgetting, because the base LM is fine-tuned on question and answer pairs from 2011. This is indeed an interesting possibility, and one that we aim to explore more in the future. We note that the exact same fine-tuning process is applied to all settings, so the 2018 results—and all the other results—are similarly obtained by fine-tuning the base LM on questions and answers from 2011; the only difference is whether the base LM is trained on data from up to 2018 or 2019 (or other years). So it could be the case that the LM that is trained on data from up to 2019 undergoes a more severe catastrophic forgetting when being fine-tuned on the 2011 question-answering dataset, although we plan to conduct further experiments to establish whether this is the case.
> * Since the paper submission, we have continued to examine whether there are any peculiarities of the 2019 data that might render the result slightly worse than training the LM only up to 2018. So far, we have not encountered any bugs or issues that might better account for this result, although we will continue to do this and update the manuscript accordingly if we discover any issues.
>
> We also thank the reviewer for the typos and presentation suggestions; rest assured we will fix them in subsequent iterations of the paper.

---

> > ### Comment · Reviewer_vYE6 · 2021-09-01
> > **Thanks for the response!**
> >
> > Glad to see that the authors continue analyzing the counter-intuitive results from Fig 4 and commit to explaining them in the next version of the paper.

---

### Author Response · Authors · 2021-08-10
**General response to all reviewers**

We thank our reviewers for their detailed comments, insights, and valuable suggestions. We are delighted that the reviewers: (i) find the paper to be thorough, coherent, and insightful (reviewers vYE6, PJPM, and bwt6), (ii) deem the underlying research question to be an important yet understudied one (reviewers vYE6 & 1qBQ), and (iii) find the recommendations, insights, and findings of the paper to be useful, both for the broader community and for future work in this area (reviewers PJPM & bwt6).

We have provided a detailed response to each reviewer, and we are happy to answer any further questions. Based on the reviewers’ valuable feedback, insights, and suggestions, we commit to the following action items:
* We will clarify further why the closed-book QA experiment is worse for the language model that is trained with data up to 2019, compared to the model that is trained with data only up to 2018 (in response to reviewer vYE6).
* We will emphasise the need for more and better benchmarks that can assess continual learning ability in important downstream NLP tasks like question answering—above and beyond the core language modelling task explored in this paper (in response to reviewer PJPM). To this end, our paper has only scratched the surface of this important research direction. In this paper, we can only consider two question-answering tasks due to space constraints, although we hope that our work will encourage the creation of more comprehensive and diverse language-based continual learning benchmarks in the near future.
* We will expand further on the potential societal impact of our work (in response to reviewer bwt6). We will argue for the importance of continually-updating language models to mitigate historical biases and prejudices, particularly concerning issues on which public perception and support have shifted over time (e.g. the #BlackLivesMatter movement). We will also emphasise the argument against the “brute-force” approach of keeping models up-to-date by periodically retraining the model from scratch, which has high computational and environmental costs given the ever-growing size of language models used in the field.
* We will include experiments assessing the extent of catastrophic forgetting after applying dynamic evaluation (in response to reviewer 1qBQ). We will also include more detailed limitations of dynamic evaluation, and outline potential avenues for improvement to make dynamic evaluation work better. This includes factorised models where only a subset of the model parameters need to be updated by dynamic evaluation (hence reducing the computational cost and improving the data efficiency of dynamic evaluation), in addition meta-learning approaches that aim to make a few steps of gradient descent as effective as possible.
We will also fix the typos, as suggested by reviewers vYE6 & PJPM.

We thank our reviewers again for their time and consideration.

---

### Decision · Program_Chairs · 2021-09-27

**Decision:**

Accept (Spotlight)

**Comment:**

This paper presents a study of temporal generalization of language models to time periods beyond which they are trained on, demonstrating that current “static” LMs do not generalize well to text from the future and degrade over time. The paper argues for studying dynamic language modeling and presents new benchmarks for the same. All the reviewers and the AC found this to be novel, solid and timely work, especially given the recent proliferation of large-scale language models in a wide variety of real-world applications.